# Sleep Quality, Nutritional Habits, and Physical Activity in Pediatric Cancer Survivors: A Dyadic Analysis Approach

**DOI:** 10.3390/nu17020250

**Published:** 2025-01-11

**Authors:** Dylan G. Serpas, Rachel Sauls, Heewon L. Gray, Marilyn Stern

**Affiliations:** 1Department of Psychology, University of South Florida, Tampa, FL 33620, USA; 2Department of Nutrition, University of North Carolina at Chapel Hill, Chapel Hill, NC 27599, USA; 3College of Public Health, University of South Florida, Tampa, FL 33620, USA; 4Department of Child and Family Studies, University of South Florida, Tampa, FL 33620, USA

**Keywords:** pediatric cancer survivors, sleep quality, nutritional habits, physical activity

## Abstract

Background/Objectives: Sleep disturbances are prevalent among pediatric cancer survivors (PCSs) and their caregivers, often leading to poorer dietary choices and reduced physical activity. Additionally, the sleep quality and health behaviors of parents and children can affect each other. This study examined bi-directional associations between PCSs and their parents’ sleep quality and health-related behaviors. Methods: 127 parent–child dyads enrolled in a behavioral intervention for pediatric cancer survivors (Mage = 11.04; 53.2% female) and their families completed the Patient Sleep Quality Inventory (PSQI) and USDA Parent or Child Food and Activity Questionnaire (FAQ). Two actor–partner interdependence models, using multi-level modeling and adjusting for relevant confounds, estimated the bidirectional associations between the parent–child dyad’s sleep quality and composite food and physical activity quality. Results: After controlling for partner BMI and actor and partner age, the effect of children’s PSQI scores on parent’s FAQ scores was statistically significant (β = −0.23, *p* = 0.036). Poorer sleep quality in children was associated with worse physical activity and dietary behaviors in parents. Second, after adjusting for race, a significant partner effect for parents’ FAQ scores on children’s PSQI scores was shown—greater parental food and physical activity quality was associated with better sleep quality in PCS (β = −0.20, *p* = 0.041). Conclusions: Better food and activity quality for parents is linked to improved sleep quality for children, while children’s poor sleep quality is associated with lower food and activity quality in parents. The findings highlight the prospective value of systems-focused clinical interventions to manage sleep quality and promote positive health behaviors among PCS.

## 1. Introduction

Pediatric cancer survivorship has steadily increased due to medical advancements, with the overall five-year survival rate for childhood cancer rising from approximately 58% in the mid-1970s to 84% in recent estimates [1]. Pediatric cancer survivors (PCSs) encounter a myriad of potential adverse health barriers, including physical limitations [2], weight gain [3], disordered eating [3], and sleep impairments [4], that may affect long-term health outcomes [5,6]. Moreover, in young adulthood, PCSs are nearly 11 times more likely to develop cardiovascular diseases than their healthy siblings, and in middle age, their likelihood of experiencing a major cardiac event increases [7,8]. Identifying and critically examining characteristics in the cancer survivorship landscape that contribute to the spectrum of physical challenges post-treatment is paramount to providing comprehensive care and support across the lifespan [5]. This underscores the incremental value of integrating a comprehensive model of health and behavior change that considers the sequelae of childhood cancer treatments for both the patients and their broader support systems, including parents and guardians.

Among their documented health barriers, sleep impairments for PCS can significantly impede their overall health and well-being, posing substantial challenges post-treatment [4]. Research has found that survivors report disruptions in their sleep patterns, including difficulties falling asleep, frequent night-time wakening, and poor global sleep quality [9,10]. Impairments in sleep quality are suggested to be a downstream consequence of various factors, including treatment-related side effects, psychological distress, and changes in circadian rhythms [4]. Moreover, the health implications of sleep loss among PCS are profound, with studies linking inadequate sleep to a greater risk of cognitive impairment, mood disorders, compromised immune function, and decreased quality of life (QOL) [3,11]. Therefore, addressing impairments in sleep quality among this population is crucial to optimizing long-term health outcomes (e.g., reducing the risk of chronic conditions) and enhancing overall QOL [9,12].

The prevalence and risks of obesity are significantly higher in PCS compared to the general population, with estimates indicating that obesity in PCS is 15% higher than national averages [13,14,15]. Modifiable lifestyle behaviors, including healthy eating and physical activity, may reduce the risk of adverse health outcomes for PCS with overweight and obesity [16]. Lifestyle behavior factors, including sleep, diet, and physical activity, are associated with the rate of obesity in this population [3,4,17]. Lifestyle interventions, particularly those that focus on nutrition and physical activity, have emerged as promising strategies for improving sleep among PCSs and families [3]. Research suggests that adopting a healthy lifestyle, including a well-balanced diet and regular physical activity, can positively influence sleep patterns and overall sleep quality [18,19]. Nutritional factors such as consuming foods rich in tryptophan, magnesium, and melatonin precursors may promote relaxation and enhance sleep onset and duration [20]. Similarly, engaging in regular physical activity may regulate circadian rhythm, reduce stress, and promote better sleep [21,22]. Additionally, involving parents in the implementation of these lifestyle interventions is essential, as they play a crucial role in facilitating healthy behaviors and creating conducive sleep environments for their children [12,23,24]. Research suggests a dyadic impact on health behaviors through parental role modeling that allows children to observe, learn, and adopt behaviors more easily and promotes long-term habit retention [25,26]. Thus, educating parents on the importance of nutrition and physical activity in sleep hygiene, encouraging family-based meal planning, and incorporating enjoyable physical activities into daily routines can empower parents to support their child’s sleep health effectively [12,27].

There are mixed findings on the bi-directional relationships regarding sleep quality, physical activity, and diet that warrant further investigation. One body of research suggests that increased physical activity is related to lower sleep duration [28], whereas other studies indicate a positive association between regular physical activity and sleep quality and duration [29]. Mixed findings warrant further exploration of this relationship among vulnerable populations, namely, PCS and their families.

### Study Purpose

This study examined bi-directional dyadic associations between sleep quality, diet quality, and physical activity in PCS with overweight and obesity and their parents. An actor–partner interdependence model (APIM) dyadic analysis approach was used to address the study objectives. Based on prior research highlighting a gap in understanding the bi-directional associations between sleep quality, diet quality, and physical activity uptake for PCS [18,26,28,29,30], we sought to examine whether parent behaviors, including sleep quality, diet, and physical activity, are positively associated with their children’s.

## 2. Materials and Methods

### 2.1. Participants and Setting

This exploratory data analysis examined cross-sectional data from a baseline sample of PCS with overweight and obesity who were recruited to participate in an intervention targeting parents of PCS to promote healthy eating, physical activity, and mindfulness behaviors in their children [31]. Data were drawn from a randomized controlled intervention for pediatric cancer survivors (Nourish-T+) [31]. Eligible participants were 5–14 years old, at least six months post-treatment, ≥85th body mass index (BMI) percentile, able to engage in physical activity tailored to their current medical status, and in remission. Eligible participants were required to be aged 5–14, since this study focused on pediatric oncology patients. Patients aged 15 to 39 are categorized as adolescents and young adults based on the National Cancer Institute’s Progress Review Group recommendations [32,33]. Participants were recruited from multiple pediatric oncology survivorship clinics and medical centers throughout the United States.

The study was conducted in accordance with the Declaration of Helsinki, and the protocol was approved by the Ethics Committee of University of South Florida (STUDY000244) on 20 May 2020.

### 2.2. Data Collection

Participants completed a demographic questionnaire involving eating and physical activity behaviors and sleep quality. Eating and physical activity behaviors were measured using a 20-item child/adult food and activity questionnaire [34] derived from the national Expanded Food and Nutrition Education program. A sum score of item responses were calculated per parent–child dyad, with higher scores indicating healthy eating and increased physical activity. Sleep was measured through the patient sleep quality inventory [35], a 19-item survey asking questions about sleep quality, sleep latency, and duration. A global score was calculated based on standard scoring procedures [35], with possible scores ranging between 0 and 21, with higher scores suggesting poorer sleep quality.

Sociodemographic variables included age in years; gender (boy or girl); race/ethnicity (White, Black, Hispanic, Asian/Pacific Islander, other); household income (>USD 20,000, USD 20,000–34,999, USD 35,000–49,999, USD 50,000–69,999, >USD 70,000); and marital status. BMI was calculated by anthropometrics (weight and height) collected by research assistants over the phone or Zoom. Medical history was collected from oncology providers, including cancer type (e.g., acute myeloid leukemia; AML), acute lymphoblastic leukemia (ALL), lymphoma, or any tumor. All demographic and medical history questionnaires were administered via the Research Electronic Data Capture (REDCap Version 14) secure online survey tool. Complete demographic characteristics are provided in Table 1.

### 2.3. Data Analysis

Descriptive statistics were computed for all primary study variables. Inter-variable correlations were calculated for children and their parents separately (see Table 2). This study sought to utilize an APIM model using distinguishable dyads; however, this choice was statistically evaluated. An Actor–Partner Interdependence Model (APIM) [36] was performed using multi-level modeling and generalized least squares with correlated errors and restricted maximum likelihood estimation [37]. All predictors were grand-mean-centered. Tests of distinguishability were conducted to assess the statistical advantages of treating the dyad as distinguishable by constraining model parameters across dyad roles and evaluating whether these constraints significantly reduced model fit. A statistically significant (*p* < 0.05) test result would indicate dyadic distinguishability such that the actor and partner effects, variances, or covariances differ significantly between dyad members. APIMs evaluated (a) how each person’s and their dyadic partner’s sleep quality were related to their own and their dyadic partner’s physical activity and nutrition quality and (b) how each person’s and their dyadic partner’s physical activity and nutrition quality were related to their own and their dyadic partner’s sleep quality. Models were screened for covariates, and all predictors were grand-mean-centered.

### 2.4. Data Screening

Data were inspected for missing values and non-normality. Due to the exploratory nature of the study, only complete cases were included in the analysis. Descriptive statistics were calculated to screen for outliers and to ensure the data met the assumptions of the APIM. Normality was assessed by screening skewness and kurtosis values. Multicollinearity was evaluated using variance inflation factors to ensure that predictor variables were not highly correlated. Linearity and homoscedasticity were evaluated through scatterplots of residuals. APIM assumptions were confirmed before analysis.

Descriptive statistics and bivariate correlations are provided in Table 1. Average scores for all study variables were not significantly different between parents and children (*p*’s > 0.05).

Univariate normality was screened by inspecting skewness and kurtosis values for continuous variables. All values were within an acceptable range of ±1. [38] No univariate (z-scores > 3.29, *p* < 0.001) or multivariate outliers (Mahalanobis distance > 18.29, *p* < 0.001) were identified for primary study variables. Linearity among primary study variables was assessed by inspecting inter-variable correlations using Pearson product–moment correlations, which is appropriate for distinguishable dyads [39,40]. Linearity was supported by moderate correlations among variables. Thus, APIM analysis proceeded.

### 2.5. Covariates

A priori covariates were screened for inclusion in the APIM analysis. The covariates evaluated for inclusion were income, sex of the child, race/ethnicity, diagnosis, BMI, and age. All covariates were screened in the analysis. Due to limited power for the proposed APIM analyses, only statistically significant (*p* < 0.05) covariates were retained in the final reported models. Covariate effects were screened to vary across parents and children such that a separate effect for parents and children was computed. Income, sex of the child, race/ethnicity, and diagnosis were inputted as between-dyad variables. BMI (kg/m^2^) and age were inputted as within-dyad variables and separate actors so separate actor and partner effects could be estimated.

## 3. Results

A total of *n* = 127 parent–child dyads (N = 254) were included in these analyses. The average age (SD) of parents/ caregivers included in this study was 40.6 (10.9) years; a majority (*n* = 90; 72%) were mothers with an average (SD) BMI of 33.9 (7.3) kg/m^2^. A little over half of the children were female (*n* = 68, 53.2%), with an average age (SD) of 11 (2.7) years. Under one-half were diagnosed with ALL (*n* = 114, 45%), and the average (SD) time since treatment for PCS was 6.2 (3) years. All remaining demographic information is provided in Table 1.

### 3.1. APIMs

Sleep Quality Effects. The model test of distinguishability for the model estimating effects of PSQI scores on FAQ scores yielded a statistically significant model chi-square, χ^2^(4) = 16.34, *p* = 0.003, indicating that constraining parameters across roles did significantly reduce model fit. In other words, parents and children could be statistically distinguished. Therefore, a distinguishable APIM was retained.

Model covariates initially included income, sex of the child, race/ethnicity, diagnosis, BMI, and age. Income, sex of child, race/ethnicity, and diagnosis were treated as between-dyad variables. BMI and age were treated as within-dyad variables such that separate actor and partner effects were screened as covariates. Covariates were screened with and without varying effects by the distinguishing variable. When varying covariates by the distinguishing variable, no covariate contributed significantly to the model (*p*’s > 0.05). When screening covariates in aggregate, partner BMI and actor and partner age contributed significantly to the model. These three covariates were retained in the final analysis. To conserve power, the remaining non-significant (*p*’s > 0.05) covariates were dropped from the analysis.

The results indicated that, after controlling for partner BMI and actor and partner age, contrary to hypotheses, actor effects for the parents’ (β = −0.14, *p* = 0.173) and children’s (β = −0.12, *p* = 0.162) PSQI scores on their own FAQ scores were not statistically significant; however, the results trended in the expected direction. As expected, the partner effect of children’s PSQI scores on parents’ FAQ scores was statistically significant (β = −0.23, *p* = 0.036) such that worse sleep quality scores of children were associated with worse physical activity and dietary behaviors in parents. The partner effect of parents’ PSQI scores on children’s FAQ scores was not statistically significant (β = −0.05, *p* = 0.534). Notably, partner BMI (β = −0.11, *p* = 0.049) and actor age (β = −0.12, *p* = 0.029) were negatively associated with sleep quality. The proportion of variance explained by the full model was approximately 10% for parents and 7% for children, which are considered small effect sizes [41].

Post hoc power analysis for APIM [33] indicated that the model examining the effects of FAQ on PSQI scores was generally underpowered to detect statistically significant effects at an alpha of 0.05. The statistically significant partner effect of children’s PSQI scores on parents’ FAQ scores was sufficiently powered (1 − β = 0.93; see Appendix A). Given the lack of power to detect all effects, caution should be used when interpreting null findings due to the increased risk of Type II errors.

### 3.2. Physical Activity and Dietary Behavior Effects

The model test of distinguishability for the model estimating the effects of FAQ scores on PSQI scores yielded a statistically non-significant model chi-square, χ2(5) = 2.61, *p* = 0.625, indicating that constraining parameters across roles did not significantly reduce model fit (i.e., parents and children were statistically similar in their effects). The null result from the test of distinguishability should be interpreted with caution, as the APIM analysis being underpowered may have increased the susceptibility of this test statistic to Type II error. In addition, the roles parents and children play in the family system are distinct, and therefore, given the exploratory nature of this study, a distinguishable APIM was retained.

Model covariates initially included income, sex of the child, race/ethnicity, diagnosis, BMI, and age, which were within-dyad variables such that separate actor and partner effects were screened as covariates. Analysis indicated that only race contributed significantly to the model. To conserve power, the remaining non-significant (*p* > 0.05) covariates were dropped from the analysis.

The results indicated, after controlling for race, in support of the hypotheses, a significant partner effect for parents’ FAQ scores on children’s PSQI scores such that greater physical activity and healthy eating of parents correlated with better sleep quality scores (indicating better sleep quality) in children (β = −0.20, *p* = 0.041). Contrary to hypotheses, actor effects for parents’ (β = −0.18, *p* = 0.060) and children’s (β = −0.07, *p* = 0.553) FAQ scores on their own PSQIs were not statistically significant; however, the results trended in the expected direction. The partner effect of children’s FAQ scores on parents’ PSQI scores was not statically significant (β = 0.02, *p* = 0.864). Notably, a significant covariate effect of race on PSQI scores was found for parents (β = 0.29, *p* < 0.001) but not children (β = 0.01, *p* = 0.896). White parents had greater PSQI scores, whereas no effect was found for children. The proportion of variance explained by the full model was approximately 10% for parents and 5% for children, which are considered small effect sizes [41].

Post hoc power analysis for APIM [42] indicated that the model examining the effects of FAQ on PSQI scores was generally underpowered to detect statistically significant effects at an alpha of 0.05. The statistically significant partner effect for parents’ FAQ scores on children’s PSQI was marginally adequately powered (1 − β = 0.75; see Appendix A). Due to insufficient statistical power to detect all effects, interpretation of null results should be approached with caution due to an increased probability of Type II errors.

Note: Results from APIM analyses are provided in Figure 1.

## 4. Discussion

This study examined bi-directional dyadic associations between PCSs’ and their mothers’ sleep quality and food and physical activity quality. This study is likely the first to examine the proposed bi-directional associations among a clinical sample of PCSs and their caregivers. The findings indicated that, consistent with the hypotheses, better parental food and activity quality was associated with better child sleep quality in children. Additionally, children’s reports of poorer sleep quality were linked to lower food and activity quality in parents. Taken together, the findings illustrate the importance of the parent–child relationship in sleep, nutrition, and physical activity in the context of pediatric oncology survivorship.

This study’s findings may offer implications for future interventions addressing the parental role in improving long-term health for pediatric cancer survivors. Meta-analytic evidence has indicated that, across 26 randomized control trials, parent-based interventions were related to improved physical activity uptake, lower screen time, and healthier eating among children compared to controls [43]. Longitudinal research has also shown that mothers’ and children’s physical activity and sedentary behavior are positively correlated during non-school time [44], highlighting the salience of the parent–child association in health behaviors [45]. Despite the evidence indicating the influential association of the parent–child relationship on health behaviors, the quality of research on family behavioral interventions warrants more robust methodology [46]. In the context of pediatric oncology, survivors’ sleep quality and physical activity are correlated [47], and patients’ sleep quality is associated with parental behavior to accommodate children’s bedtime requests [48]. Family-based interventions to promote physical activity [49], nutrition [50,51,52], and sleep [53] exist and demonstrate efficacy; however, interventions that target this constellation of health behaviors are less common [31].

### Limitations and Future Directions

This study is not without limitations. First, this study employed a cross-sectional design to analyze baseline data of an ongoing behavioral intervention for caregivers of pediatric cancer survivors. The cross-sectional study design precludes the ability to interpret the temporal stability and causal nature of the associations examined in this study. Future research is needed to investigate the cross-lagged associations of sleep quality, physical activity, and dietary behaviors among PCSs and their caregivers. Cross-lagged prospective and ecological momentary assessment research designs may be leveraged to examine the longitudinal associations of the variables. This may clarify the complexities that lie within these associations. This study was also generally underpowered to detect all possible effects. The increased risk of Type II error in this study indicates that null findings should be interpreted with caution, and future research is needed to replicate these findings in sufficiently powered samples. However, the statistically significant effects that were detected obtained adequate power and are, thus, likely reliable observations. In addition, data on sleep quality, nutritional habits, and physical activity were obtained via self-report, which is vulnerable to response bias. Future research is needed to objectively assess these constructs and verify the associations under study. Substantial evidence linking sleep quality and duration with diet remains cross-sectional, and although existing evidence suggests that a bi-directional link with improved diet quality improves sleep quality and duration [30,54], this lacks interventional investigation in well-controlled efficacy trials. Efficacy trials that leverage family systems to modulate behavior change may be useful in improving sleep quality, nutrition, and physical activity for PCS as a means of reducing their risk of adverse health outcomes in adulthood. Lastly, potential mechanisms that underlie associations among sleep quality, physical activity, and nutrition are critical. For instance, PCSs’ family systems contain risk and protective factors that may explain health behaviors, including family cohesion, ritual, environment, functioning, and parental stress [55]. Potential mediating factors could be integrated into efficacy trials to identify factors that impede or maintain positive health behaviors.

## 5. Conclusions

This exploratory cross-sectional study provides evidence for the bi-directional associations among parental and children’s food, activity, and sleep quality, underscoring the interconnected associations among these constructs. The findings provide preliminary evidence that could inform targeted psychosocial and behavioral interventions for pediatric survivors and their families. Healthcare providers may consider incorporating these observations to encourage families to adopt integrated approaches to improving dietary habits, physical activity, and sleep routines. For example, establishing and maintaining consistent family rules and routines around sleep, physical activity, screen time, and meals—which may be disrupted throughout the cancer process [56]—may benefit overall family well-being. Future research should build on and leverage this evidence to develop clinical practice guidelines and intervention strategies tailored for PCSs and their families.

## Figures and Tables

**Figure 1 nutrients-17-00250-f001:**
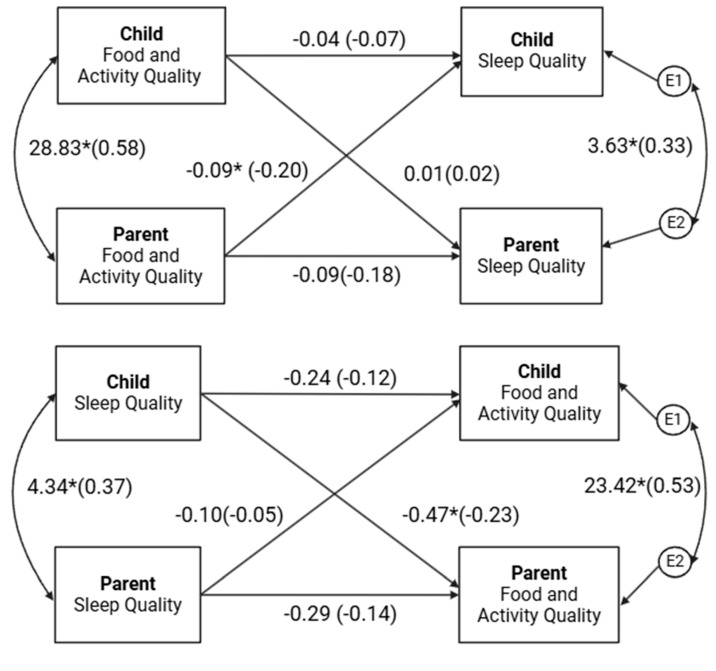
Actor–Partner Interdependence Models. Note: The results display the dyadic bi-directional associations between parents’ and children’s sleep quality and food and activity quality. Models were estimated using actor–partner interdependence models, which account for actor effects (associations between an individual’s own variables) and partner effects (associations between one dyad member’s variable and the other member’s variable). Higher sleep quality scores suggest worse sleep quality. * *p* < 0.05.

**Table 1 nutrients-17-00250-t001:** Demographics of parent–child groups for study sample (*n* = 254).

	Parent (*n* = 127)	Child (*n* = 127)
Age (mean ± SD) years	43.2 ± 7.6	11.0 ± 2.7
^a^ BMI (mean ± SD) kg/m^2^	33.4 ± 9.3	34.4 ± 8.72
^a^ BMI Percentile (mean ± SD)	--	95.6 ± 2.2
Gender		
Female: *n* (%)	90 (72.3)	67 (52.8)
Race/Ethnicity: *n* (%)		
White	51 (40.2)	51 (40.2)
Black/African American	23 (18.1)	23 (18.1)
Hispanic/Latine	44 (34.6)	44 (34.6)
Asian/ Pacific Islander	4 (3.1)	4 (3.1)
Other	2 (1.6)	2 (1.6)
Missing	3 (2.4)	3 (2.4)
Cancer Diagnosis		
^b^ ALL	--	57 (44.9)
Tumor	--	31 (24.4)
^c^ MDS	--	1 (0.8)
^d^ AML	--	11 (8.7)
Missing	--	27 (21.3)
Time since Diagnosis	--	6.2 ± 3.0
Income: *n* (%)		
<USD 20 K	25 (19.7)	--
USD 20–34,999	24 (18.9)	--
USD 35–49,999	13 (10.2)	--
USD 50–69,999	11 (8.7)	--
>USD 70 K	52 (40.9)	--
Missing	2 (1.6)	--

^a^ BMI = body mass index; ^b^ ALL = Acute Lymphoblastic Leukemia, ^c^ MDS = myelodysplastic syndromes, ^d^ AML = acute myeloid leukemia.

**Table 2 nutrients-17-00250-t002:** Bivariate correlations.

Variable	1	2	3	4	5	6	7	8
1. Parent FAQ	-							
2. Child FAQ	0.576 **	-						
3. Parent PSQI	−0.201 *	−0.104	-					
4. Child PSQI	−0.250 **	−0.194 *	0.357 **	-				
5. Child age	−0.159	−0.236 *	0.032	0.063	-			
6. Parent Age	−0.101	−0.172	−0.031	0.090	0.195 *	-		
7. Parent BMI	−0.166	−0.189 *	0.099	0.128	0.614 **	0.005	-	
8. Child BMI	−0.193 *	−0.066	0.061	0.135	−0.045	0.002	0.084	-

Note: FAQ = Food and Activity Quality; PSQI = Patient Sleep Quality Index; BMI = body mass index. (*n* = 127). * *p* < *0*.05, ** *p* < 0.001.

## Data Availability

The NOURISH-T+ study is ongoing, and we are still in the process of collecting project data. Final data will be made available once the project is completed. Data used for this article will be shared upon request, and a data use agreement will be established. Upon study competition, we will comply with the National Cancer Institute data-sharing policies and submit data as required.

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
