# Peer review of "Sleep Quality, Nutritional Habits, and Physical Activity in Pediatric Cancer Survivors: A Dyadic Analysis Approach"

_nutrients, 2025, doi:10.3390/nu17020250_

Round 1

Reviewer 1 Report

Comments and Suggestions for Authors

Peer-review report for the article "Sleep Quality, Nutritional Habits, and Physical Activity in Pediatric Cancer Survivors: A Dyadic Analysis Approach."

Overall Evaluation

The study addresses an important topic of bi-directional relationships between sleep quality, dietary habits, and physical activity in pediatric cancer survivors and their parents. The dyadic approach is novel, particularly in the context of pediatric cancer survivorship. However, there are areas where the clarity, methodology, and implications could be strengthened.

Abstract

Strengths:

  • Clearly summarizes the study objectives, methods, and key findings.
  • Highlights the bidirectional nature of the parent-child relationship.
  • Emphasizes the relevance of systems-focused approaches.

Points for Improvement:

  • The abstract lacks specificity regarding the methodologies, e.g., the inclusion of covariates or model specifics (Actor-Partner Interdependence Model, APIM).
  • More explicit mention of the implications of findings for clinical interventions would be beneficial.

Introduction

Strengths:

  • Provides a comprehensive background on the challenges faced by pediatric cancer survivors, emphasizing the significance of sleep, diet, and physical activity.
  • Identifies research gaps effectively and justifies the dyadic analysis approach.

Points for Improvement:

  • The hypotheses could be more explicitly linked to the gaps highlighted in the introduction.
  • Incorporate a stronger discussion on why the chosen age range (5-14 years) is particularly relevant to the study.

Methods

Strengths:

  • Detailed explanation of the study design, recruitment, and eligibility criteria.
  • Clear description of the instruments used for measuring sleep quality (PSQI) and dietary/physical activity behaviors (FAQ).

Points for Improvement:

  • The statistical power limitations were acknowledged, but a clearer explanation of how these limitations impact the validity of the findings is necessary.
  • Consider addressing potential biases in using self-reported data for FAQ and PSQI measurements.
  • While the APIM was employed, the justification for its distinguishability testing results could be expanded for clarity.

Results

Strengths:

  • Presents clear and structured findings, with an emphasis on the significant partner effects.
  • Descriptive statistics provide a detailed view of the sample characteristics.

Points for Improvement:

  • The results section could better integrate effect sizes to contextualize the findings' practical significance.
  • Provide more explanation for the underpowered nature of certain analyses and its implications for interpreting non-significant findings.
  • The inclusion of visual aids like Figure 1 is helpful but could be more descriptive to aid in interpretation.

Discussion

Strengths:

  • Effectively connects the results with existing literature, emphasizing the role of parent-child dynamics.
  • Suggests practical applications for interventions aimed at improving lifestyle factors in pediatric cancer survivors.

Points for Improvement:

  • Address the limitations of the cross-sectional design more explicitly, especially the inability to infer causation.
  • Expand on potential mechanisms underlying the bidirectional associations observed.
  • Suggestions for future research could be more detailed, especially concerning longitudinal studies or intervention designs.

Statistical Analysis

Strengths:

  • The use of APIM is appropriate for the dyadic nature of the research question.
  • Covariates like age, BMI, and race were appropriately screened and included in the models.

Points for Improvement:

  • The statistical assumptions and their tests (e.g., multicollinearity, normality) were mentioned but could be elaborated upon to reassure the reader of the robustness of the analyses.
  • Address potential model misfit issues and the implications of retaining distinguishable APIM despite non-significant chi-square results in one model.

Conclusion

Strengths:

  • Provides a concise summary of findings and their implications.
  • Reinforces the importance of parent-focused interventions.

Points for Improvement:

  • Include a stronger take-home message that underscores the practical implications of the findings for clinical settings.
  • Consider elaborating on how these findings can be translated into action-oriented guidelines for families and healthcare providers.

Positive Aspects of the Article

  1. Novel dyadic approach, highlighting the interplay between parents and children.
  2. Focus on a vulnerable population, with actionable implications for health interventions.
  3. Comprehensive use of validated tools for assessing key variables.

Key Areas for Improvement

  1. Clarity and Specificity: Expand on statistical methodology and interpretative frameworks.
  2. Limitations: Address limitations in a more detailed manner and propose strategies to mitigate them in future studies.
  3. Practical Implications: Enhance discussion of how findings can influence clinical practices and intervention designs.
  4. Power Analysis: Provide a clearer explanation of post-hoc power issues and their implications for reliability.

Reviewer 2 Report

Comments and Suggestions for Authors

Thank you for allowing me to read this work - an interesting topic and an extremely important field of research

In lines 84-90 it is proposed to withdraw from the hypotheses and introduce the purpose of the work

In the methodological part, I suggest writing a flow chart for the selection and course of research - it will be more transparent.

The discussion should be expanded to indicate the strengths and weaknesses of the study and practical implications. I provided detailed information in the comments in the text.
